# Identifying pragmatic solutions to reduce cigarette smoking prevalence in Indigenous North Americans: A sequential exploratory mixed-methods study protocol

**Ann M. Rusk**[1,2,3,4,5]*, **Maggie Paul**[3], **Dan P. Kelleher**[3], **Jon Tilburt**[6,7], **Donald Northfelt**[8,9], **Matthew Rank**[3,6,10,11,12], **Rodrigo Cartin-Ceba**[1,2], **Guthrie Capossela**[13], **Trudie Jackson**[14], **Corinna Sabaque**[13], **Alanna M. Chamberlain**[15,16], **Victor E. Ortega**[1,7], **Roberto Benzo**[17], **Cassie Kennedy**[5,17]

1 Division of Pulmonary Medicine, Mayo Clinic, Phoenix, Arizona, United States of America, 2 Department of Critical Care Medicine, Mayo Clinic, Phoenix, Arizona, United States of America, 3 Robert D. and Patricia E. Kern Center for the Science of Health Care Delivery, Mayo Clinic, Rochester, Minnesota, United States of America, 4 Robert A. Winn Diversity in Clinical Trials Award Program, United States of America, 5 Respiratory Health Equity Clinical Research Laboratory at Mayo Clinic, Rochester, Minnesota, United States of America, 6 Mayo Clinic Department of Internal Medicine, United States of America, 7 Department of Quantitative Health Sciences, Mayo Clinic, Phoenix, Arizona, United States of America, 8 Department of Hematology and Oncology, Mayo Clinic, Arizona, United States of America, 9 Phoenix Indian Medical Center, United States of America, 10 Division of Allergy and Immunology, Mayo Clinic, Scottsdale, Arizona, United States of America, 11 Department of Head and Neck Surgery, Mayo Clinic, Phoenix, Arizona, United States of America, 12 Division of Pulmonology, Phoenix Children's Hospital, Phoenix, Arizona, United States of America, 13 Mayo Clinic, Rochester, Minnesota, United States of America, 14 Mayo Clinic Center for Health Equity and Community Engagement Research, Phoenix, Arizona, United States of America, 15 Department of Quantitative Health Sciences, Mayo Clinic, Rochester, Minnesota, United States of America, 16 Department of Cardiovascular Medicine, Mayo Clinic, Rochester, Minnesota, United States of America, 17 Division of Pulmonary, Critical Care, and Sleep Medicine, Mayo Clinic, Rochester, Minnesota, United States of America

* rusk.ann@mayo.edu

**Data Availability Statement:** No datasets were generated or analysed during the current study. All

## Abstract

### Background

American Indians and Alaska Natives (AI/AN) have the highest prevalence of cigarette smoking of any race or ethnicity in the United States. Efforts to address smoking prevalence in this population have not historically targeted maintenance of smoking cessation, or behaviors associated with pregnancy. Recent longitudinal cohort studies have identified maintenance of cessation and pregnancy as potential opportunities to address smoking in AI/AN people.

### Methods

To promote success in achieving sustained smoking cessation in AI/AN people, we propose a community engaged sequential exploratory mixed-methods study focused on identifying pragmatic elements of cessation interventions. A discovery sample of 45 AI/AN people will be included in the qualitative study in one of two arms consisting of small groups or one-on-one interviews to develop elements of cessation interventions for evaluation in a discrete

relevant data from this study will be made available upon study completion.

**Funding:** This study has been approved by the Mayo Clinic IRB, (number 23-011852) on 12/11/ 2023. Enrollment will occur on a rolling basis until completed for up to two years.

**Competing interests:** The authors have declared that no competing interests exist.

choice experiment survey. These one-on-one interviews will characterize the key drivers of smoking relapse and unique experiences of smoking during pregnancy. An additional, independent small group will consist of counselors who engage in smoking cessation counseling. A larger-scale survey will be administered to an AI/AN cohort from Olmsted County, Minnesota (n = 898). Elements of successful interventions will be used to inform a smoking cessation intervention pilot study. Community stakeholders have informed the methods outlined in this protocol, and there is a longitudinal engagement plan for the duration of study.

## Discussion

We outline the methods to understand optimal strategies to promote sustained cigarette smoking cessation and cessation during pregnancy in AI/AN people. This study is critical to inform a pilot intervention aimed at reducing smoking prevalence in AI/AN people.

## Introduction

Indigenous North Americans, or American Indians and Alaska Natives (AI/AN) have the highest prevalence of cigarette smoking of any race or ethnicity in the United States (U.S.), with an estimated 34.9% of adults using smoked tobacco products [1]. Smoking directly contributes to the risk for pulmonary disease, cardiovascular disease, cancer, and exacerbates disparities in Indigenous health [2, 3]. Among AI/AN people, there is wide variation in cigarette smoking prevalence among different Tribes and geographic regions due to diverse cultural practices, beliefs, and traditions [4]. There are 574 federally recognized Tribes in the United States, with more that are not federally recognized, living in urban, rural, and Reservation lands [5]. The cultural role of tobacco, access to cigarettes, advertising practices, tobacco regulation and smoking bans, all vary between Sovereign Tribal Nations, resulting in variable smoking prevalence [6, 7]. The Southwest Tribes have a lower prevalence of cigarette smoking reported compared to Great Lakes Tribes or Great Plains Tribes, with a prevalence between 10–19% to as high as 59%, respectively [6, 8, 9].

A longitudinal study of Indigenous people living in Olmsted County, Minnesota (n = 898) from 2006–2019 identified in the Rochester Epidemiology Project (REP) [10] demonstrates AI/AN individuals were more successful at achieving smoking cessation than non-AI/AN people (relative risk [RR], 1.10; 95% CI, 1.06 to 1.13; P < .001), but were >3 times more likely to resume smoking after cessation even after adjusting for an area-level measure of socioeconomic status incorporating housing and zip code indices (RR, 3.03; 95% CI, 2.93 to 3.14; P < .001) [7, 10–14]. Pregnancy events identified in the same cohort revealed AI/AN people were less likely to quit than non-Indigenous despite the equitable distribution of smoking cessation interventions (odds ratio [OR]: 0.23, 95% CI = 0.07–0.72, p = .012) [15]. It is of critical importance to identify barriers to cessation as cigarette smoking directly increases mortality in AI/ AN pregnant people [16–20]. These behaviors have not been previously described and demonstrate unique opportunities to address smoking prevalence in AI/AN people.

Prior attempts to promote smoking cessation in AI/AN people have had limited success [21, 22]. Elements associated with achieving cessation may include utilization of individualized cessation counseling sessions, planning and implementation of interventions including input from Tribal partners, and access to evidence based interventions including pharmacotherapy [22]. There are no interventions published attempting to address barriers to maintenance of

smoking cessation in AI/AN people, and successful smoking cessation interventions during pregnancy in AI/AN people have not been identified [22].

Understanding motivations to quit smoking, causes of relapse, and the unique sociocultural stressors caused by colonization and racism influencing AI/AN people is needed to inform an effective intervention for AI/AN people [22–26]. This study aims to describe elements of successful strategies to promote sustained cessation and cessation during pregnancy among AI/AN people to inform a pilot intervention. With qualitative methods, we will identify barriers to cessation in patients of childbearing age, and describe interventions that are desirable, accessible, and effective using a community-engaged approach. Qualitative results will be used to inform a survey to understand optimal elements of a smoking cessation pilot intervention. We hypothesize there are unique historical, cultural, and equity factors impeding sustained smoking cessation among AI/AN people and that mixed methods data will enable the development of successful interventional strategies.

## Materials and methods

### Study design

This is a sequential exploratory mixed methods study utilizing both qualitative and quantitative research methods. Qualitative data will be used to develop a discrete choice experiment (DCE) survey to inform a future pilot intervention with a "pick the winner" phase II randomized control trial design to identify optimal cessation program elements [27].

### Aims

The aims of this study are to 1) identify barriers to long-term maintenance of smoking cessation and cessation during pregnancy among self-identified AI/AN individuals with one-on-one interviews including themes surrounding traditional tobacco, alternative smoking methods such as electronic cigarettes, and the role of historical trauma and 2) evaluate pragmatic elements of smoking cessation interventions including accessibility and motivation with iterative, small-group qualitative interviews utilizing Indigenous storytelling methods including talking circles, AI/AN facilitated discussion, and an opening prayer with self-identified AI/AN people who formerly or currently use cigarettes, and professionals engaged in tobacco cessation counseling of AI/AN people to inform a questionnaire utilizing DCE methodology [28].

### Indigenous research considerations and community engagement plan

The designation "American Indian" or "Alaska Native" are not only racial and ethnic terms, but also represent a political identity. AI/AN people who may or may not be enrolled in or eligible for Tribal enrollment are eligible for this study to be inclusive of AI/AN people who may not have federal enrollment status. Sovereignty in the context of clinical research requires acknowledgement that data collected, and research conducted on Tribally governed lands is to be owned and approved by local governance procedures including but not limited to Tribal Institutional Review Boards (IRB), Community Research Boards (CRB), Community Advisory Boards (CAB), local, and state laws [29]. Although there are not specific IRB processes for Urban AI/AN people residing away from their geographic community, the research team is committed to ethical and equitable research with 100% compliance in coordination with work conducted with Tribal members by conducting community engaged research with iterative feedback through multiple venues [5]. The research team understands any data collected with sovereign Tribal members is not owned by the research team. Any data sharing requests will

comply with Tribal, State, and Federal data regulation procedures before release to external entities.

The conduct of any research including members of Sovereign Tribal Nations requires direct feedback, input, and collaboration with the people participating in the study [5, 29]. The principal investigator (PI) created this protocol in consultation with the Mayo Clinic Healthy Nations Advisory Board, and input from Community Engagement (CE) Studios hosted in Phoenix and Minnesota consisting of individuals who are self-identified AI/AN people with former or current smoked tobacco use. The CE Studio creates a framework for stakeholders to provide immediate feedback to the researcher on specific areas of concern before the research project is implemented [30]. Additional CE Studios will be hosted for interpretation of results and crafting the DCE survey. The PI will present progress at 6–12-month intervals as recommended by the Mayo Clinic Healthy Nations Advisory Board. This study was presented in its entirety by the PI to the Healthy Nations Advisory Board on 9/26/2023 with feedback incorporated after return of written feedback on 10/10/2023, with a plan to return no later than one year with updates. This project was also presented to Arizona CE Studio participants on 10/26/2023, and Minnesota CE Studio participants on 11/10/2023. In addition to community member input and recruitment, there will be ongoing collaboration and input from the Mayo Clinic Center for Health Equity and Community Engagement Research in Rochester, MN, and Phoenix, AZ. This project was approved by the Mayo Clinic IRB (number 23–011852) on 12/11/2023.

## Consent and recruitment

All subjects must provide consent. Methods of consent include a verbal consent script to be completed with all participants in addition to review of electronic version of consent script. After reviewing the electronic version of consent and verbal consent, subjects are again read the verbal consent script at time of qualitative interview. Subjects may decline participation at any time. Pediatric individuals (individuals under age 18) require parental consent in addition to individual assent. For the survey portion of the study, individuals may provide consent electronically, by paper, or over the phone. A copy of IRB approved consent materials are included in S1 Appendix.

Subjects will be recruited using approved flyers placed at community events attended by the Office of Mayo Clinic Community Engagement including health fairs, celebrations including pow wows, community clinics, community events, and community members who interact with peers for "word of mouth" dissemination. These methods were identified as optimal recruitment targets through community engagement studio events in Arizona and Rochester. Participants will be recruited from Minnesota and Arizona for a goal of equal enrollment. The urban AI/AN population in Phoenix, Arizona represents one of the largest urban Native groups in the United States. Of the 9.7 million AI/AN in the United States, an estimated 78% live outside of the 324 federally recognized Reservation Lands [31]. It is critical to include more than one region when evaluating Indigenous populations due to the diversity of Tribes across the county. In addition to including the Southwest, Minnesota represents a large AI/AN population with cultural values and practices unique to the Great Lakes region of the US.

All subjects will be paid one time after completing their interviews in the amount of $50 in the form of a virtual cash card. AI/AN participants will also be given a culturally appropriate gift such as sweetgrass, sage, cedar, or dried willow bark. Additionally, participants who are smokers will be offered an optional nicotine replacement product such as nicotine patches, gum, or lozenges after completion of the interview if desired by the participant. For minors under age 18 participating with parent or guardian permission, payment will be issued to the parent or guardian.

## Study participants and enrollment strategies

Participants enrolled in the qualitative study include individuals aged 14 years or older who self-identify as AI/AN who are current or former smokers, and smoking cessation counselors. The enrollment strategy for AI/AN participants with inclusion and exclusion criteria can be found in Fig 1 titled Qualitative study enrollment, American Indian and Alaska Native participants. The enrollment strategy with inclusion and exclusion criteria for small group interviews with cessation counselors can be found in Fig 2 titled Qualitative study enrollment strategy. A total of 45 AI/AN individuals will be recruited, screened, and assigned to qualitative arm 1, consisting of two small groups, or arm 2, consisting of individual interviews. To protect the privacy of pediatric patients, those under age 18 who have parental consent and who provide assent to participate will only be eligible for one on one interviews. Given the nature of qualitative data, this study is unblinded. This study was approved by the Mayo Clinic Institutional Review Board, #23–011852, on December 11[th], 2023. Enrollment will tentatively begin in January 2024 and conclude by December 31, 2024.

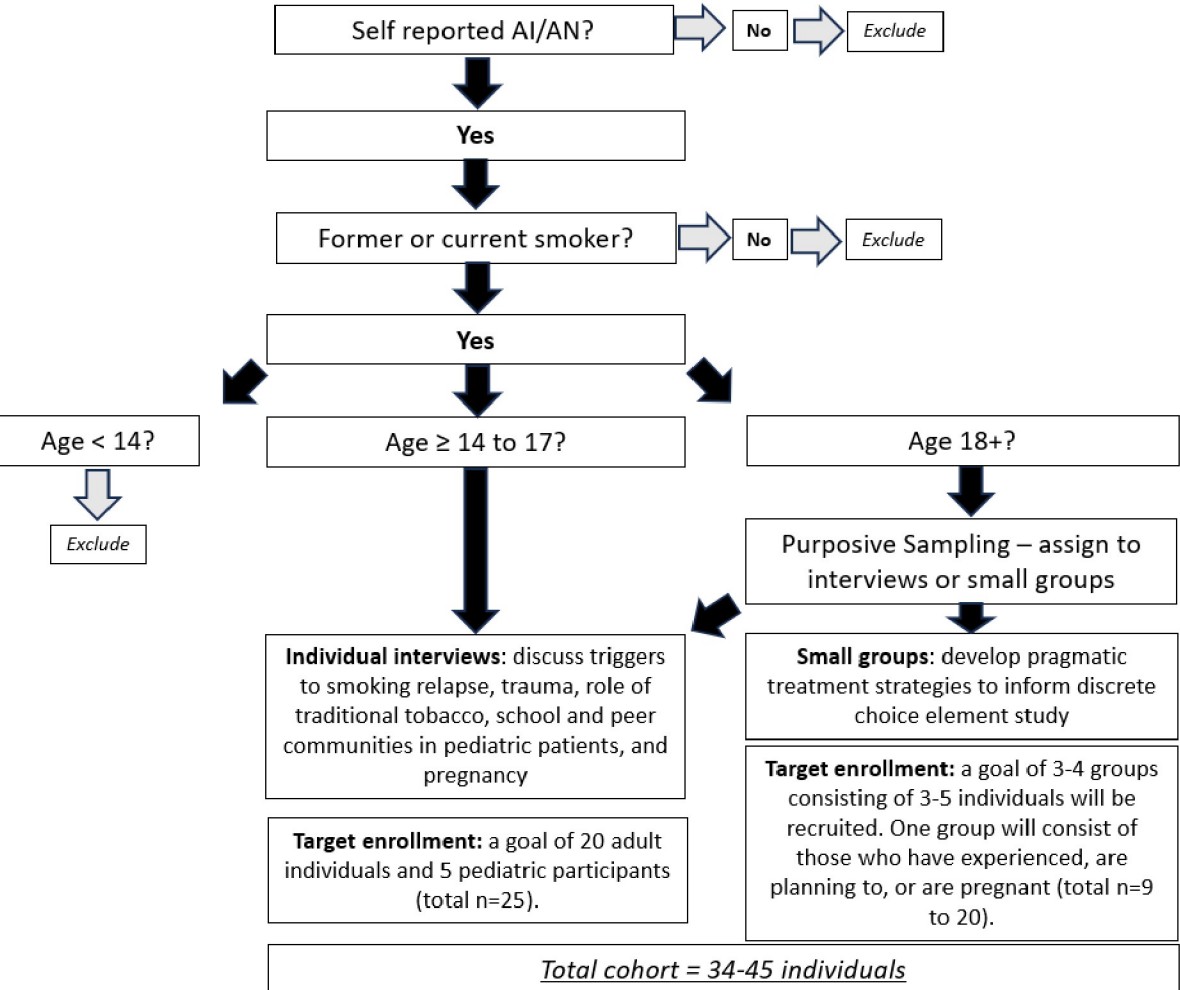

**Fig 1. Qualitative study enrollment strategy, American Indian or Alaska Native participants.** Inclusion and exclusion criteria and cohort allocation for study enrollment.

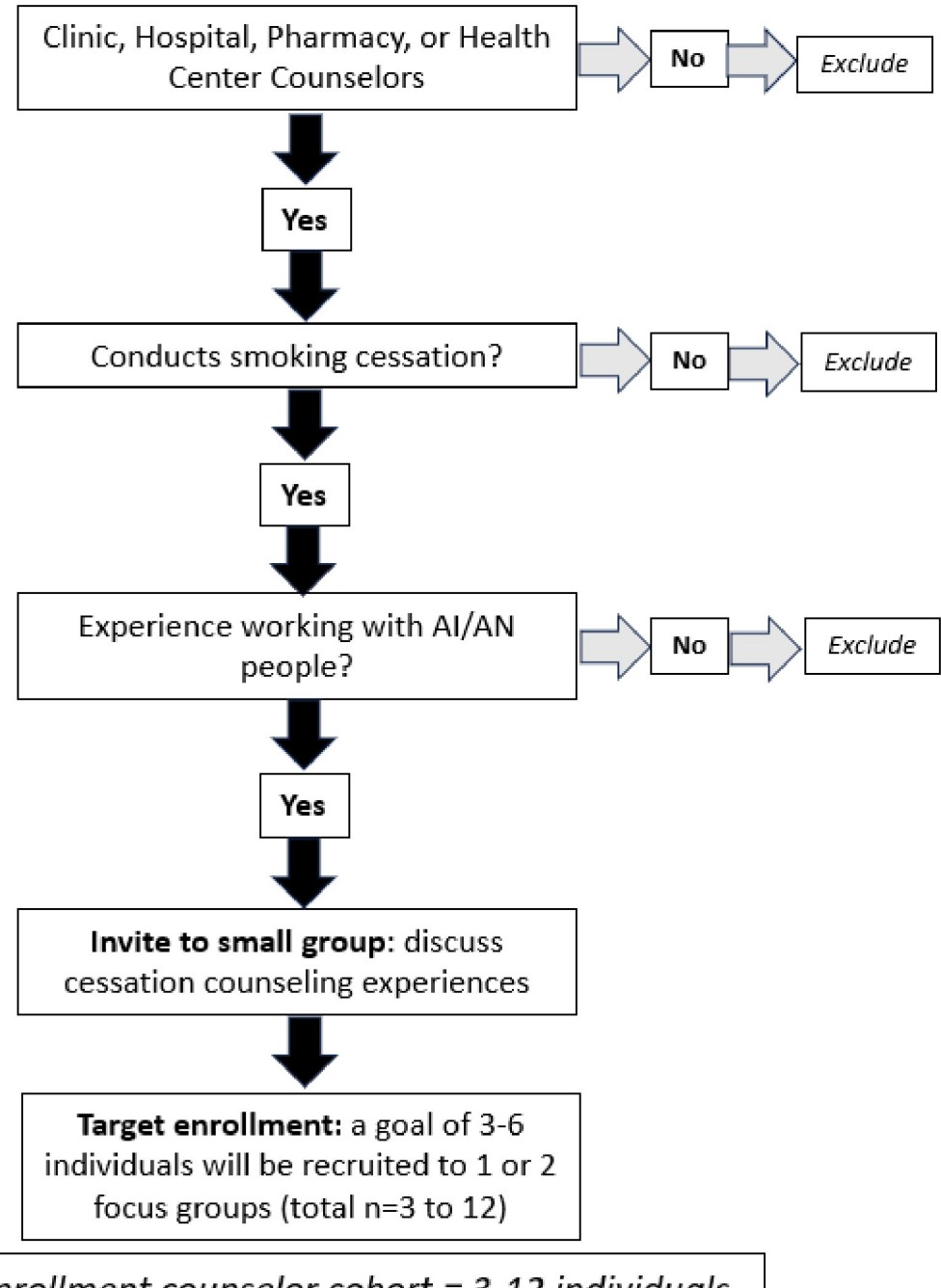

**Fig 2. Qualitative study enrollment strategy.** Inclusion and exclusion criteria for smoking cessation counselor cohort.

**Qualitative arm 1.** The goal of arm 1 is to discuss treatment elements of smoking cessation interventions and maintenance of cessation interventions to inform the design of the survey. We will recruit adult AI/AN persons who are current or former smokers to participate in four or more small group in-person discussions consisting of 3–5 individuals to explore

treatment-oriented solutions to smoking relapse, and cessation during pregnancy. One small group will consist entirely of individuals who have experienced pregnancy, are currently pregnant, or are planning pregnancy. Another small group will consist of professionals engaged in tobacco cessation counseling to understand resources, successes, and challenges that exist in smoking cessation counseling (Fig 2, smoking cessation counselor group). Smoking cessation professionals will include any individual working in a healthcare delivery setting with professional backgrounds including community health workers, pharmacists and pharmacy technicians, mental health professionals, registered nurses, medical assistants, respiratory therapists, or public health professionals. A DCE is a survey-based approach that elicits the preferences of individuals by asking them to make trade-offs between different elements of a healthcare intervention, such as a smoking cessation intervention program [32]. In the design of DCE studies, best practice guidelines recommend conducting qualitative research with a group of participants intended to be recruited into the DCE survey to understand what elements of a given a healthcare intervention are important to them [33]. When conducting qualitative research prior to the design of a DCE survey, the aim of such research is to inform the selection of attributes (elements) to be included in the DCE survey that are both feasible to implement in existing systems, and of interest and efficacy to end users. As such, DCE studies then allow for the identification of attributes that are most important to end users, allows iterative revisions of treatment choices informed by qualitative methods to guide real world treatment plans, and allows tailoring of cessation strategies that may be applied in limited resource settings [34, 35]. Including the knowledge of existing systems by incorporating community stakeholders and cessation counselors in DCE development and interpretation of results is critical to create a feasible intervention. This strategy is ideal to understand how a pilot intervention implemented in clinics serving AI/AN people can tailor their existing resources to optimize smoking cessation outcomes in their community. An example template for iteration one of the DCE survey is included in S2 Appendix. The results of this study will result in an IRB addendum to launch the finalized DCE survey to AI/AN people residing in Olmsted County, Minnesota (n = 898) identified in previous study [7, 15].

**Qualitative arm 2.** The goal of arm 2 is to understand smoking triggers, barriers to cessation, understand traditional tobacco and the use of alternative delivery methods such as electronic cigarettes, and explore novel interventions to promote cessation. We will conduct up to 25 one-on-one semi-structured interviews of adult (age 18 years and older) and minor (age 14–17 years) participants via phone, video call, or in-person, utilizing multiple methodologies, including grounded theory, and participatory action research with multimedia elements including photo storytelling elements in which a participant may use a picture to help describe their story or experiences. The PI will lead all arm 2 interviews and communicate with methods that are respectful to AI/AN people. To contextualize our discussion, we will begin each interview by providing a brief summary of prior literature demonstrating AI/AN people are resilient, motivated, and successful in quitting smoking with the additional finding revealing high risk for smoking relapse and difficulties quitting smoking while pregnant [7, 15]. Additional themes explored during interviews will include how wellness and health are viewed, and the relationship between the role of traditional tobacco use and alternative nicotine delivery devices such as electronic cigarettes.

## Data analysis

Primary endpoints include saturation of two key study objectives, including strategies to promote sustained cessation and barriers to cessation during pregnancy. Interviews will continue to the target enrollment of 45 subjects unless coding of qualitative data does not reveal new

themes after 2 or more interviews, or there are no iterative changes recommended to the DCE survey. Secondary endpoints include the evaluation of themes surrounding wellness, well-being, and how traditional and ceremonial tobacco is viewed in the lens of health in AI/AN people. Additional secondary themes surrounding tobacco use will be reviewed, including the use of alternative tobacco delivery methods, including but not limited to traditional tobacco, electronic cigarettes, vaping devices, smokeless tobacco, cigars, cigarillos, and pipes. All data will be reviewed on a continuous basis for the duration of the study.

All audio data will be recorded using a Sony-PX Series Digital Voice Recorder with upload of recordings to NVivo software for coding and thematic analysis. Audio data will be password protected and encrypted when stored on Mayo Clinic servers. Following transcription, audio data will be deleted. Additional transcription technology may include Otter voice transcription. There will be no access to health data, electronic medical records, or other health related information collected from this study. Demographic information including age, sex, cultural connection, tobacco use history, and pregnancy data, when applicable, will be collected from interviews. This information will be stored on a secure server with password protection, and will only be available to study staff. Additional notes will be collected by the PI during interviews, with notes destroyed after conversion to electronic data.

A survey defining smoking cessation program elements highly likely to be successful will be iteratively developed based on small group sessions. A survey including no more than 12 questions will be developed with the goal to identify key elements end users find as useful components of a comprehensive smoking cessation program that is sustainable to implement in a wide variety of community settings. The survey will include cessation elements identified through qualitative interviews that are both readily available in existing community infrastructure, and of interest to end users. The perspective of smoking cessation counselors who regularly work with AI/AN people is critical to understand what resources are feasible to include in ideal cessation interventions. The survey will consist of participants choosing between two cessation interventions consisting of multiple attributes, or choosing neither option. The attributes of cessation programs will be iteratively developed and pretested through a combination of qualitative focus groups, and community engagement events including CE Studios [36]. Survey will be mailed with two options to complete, on paper or via QR code scan. Results will be loaded into Qualtrics for analysis and review. A survey target response rate of 30% of the REP cohort will be the goal (goal N = 270), with additional mailings completed to non-responders.

## Dissemination plan

Data will be presented to Community Advisory Boards (CABs) recruited from CE studios, Sovereign Tribal Nations who choose to participate (if applicable), and through presentation at one year with the Healthy Nations Advisory Board. Feedback will be solicited to inform need for additional analyses, interpret results with local expertise and create framework for how to share results with the Minnesota Native Americans and Phoenix area Native Americans. Authorship will be offered to leadership of governing bodies (including CAB members, the Healthy Nations Advisory Board, and Tribal Nations who choose to participate) who would like to contribute to manuscript and abstract development. If authorship is not applicable, acknowledgement will be provided in manuscripts and presentations. All drafts of infographics, abstracts, manuscripts, and other presentations of data will be shared with governing bodies prior to submission or presentation, and data will only be presented with input from governing bodies.

## Discussion

Smoking cessation remains one of the most important elements of public health programming [37]. Despite concerted efforts to address smoking cessation, smoking prevalence remains disproportionately high among AI/AN people without effective interventions [22]. In the context of AI/AN health, cessation efforts require acknowledgement of the context of the cultural and historic roots of tobacco as a medicine. Addressing smoking prevalence requires a multifaceted approach but acknowledges the cultural significance of tobacco while implementing sustainable interventions rooted in community engagement [22, 38].

This study is unique because it not only aims to describe the drivers of smoking prevalence, but also to define the most effective existing solutions. By identifying solutions that are effective, interventions may be focused on feasible strategies that consider resource availability, community infrastructure, and healthcare systems. This pragmatic approach ensures interventions are not only culturally appropriate, but practical and implementable. Understanding the limitations of implementing interventions in existing complex models of care represents a limitation of this study. Although these results will be helpful to create a cessation intervention, they may not be sustainable, or generalizable to AI/AN populations outside of those included in the study. A pilot to understand the validity of these results will be necessary to determine if results are appropriate and can be scaled to a community level intervention.

Engaging the community is critical when designing effective and culturally tailored smoking cessation programs [39]. The history and traditions surrounding tobacco within diverse AI/AN cultural practices necessitates an approach that is respectful and integrates cultural nuance. The diversity of practices among tribal communities presents a possible limitation for findings. Effective measures in one community may not seamlessly translate to another due to distinct cultural nuances and beliefs surrounding tobacco use. Although this may present a significant challenge, ongoing study in a community engaged fashion can allow an adaptable program to be developed for diverse settings.

## Conclusions

This study will add to the limited literature describing components of smoking cessation interventions that are designed to promote prolonged, sustained cessation in American Indian and Alaska native people. This study is unique as it has community engaged methodology from inception to dissemination. With these results, we will create a pilot intervention with elements outlined as useful from community members informed by the needs of tribal communities.

## Supporting information

**S1 Appendix. Includes IRB approved consent materials.**
(DOCX)

**S2 Appendix. Template of discrete choice experiment survey to be developed with qualitative data.**
(DOCX)

## Acknowledgments

The authors would like to thank the ongoing input from community members who have participated in Community Engagement Studios, advice from the Healthy Nations Advisory Board, and the input from the Mayo Clinic Centers for Health Equity and Community Engagement Research in Rochester, MN, and Phoenix, AZ, for the support of this study.

## Author Contributions

**Conceptualization:** Ann M. Rusk, Guthrie Capossela, Alanna M. Chamberlain, Roberto Benzo, Cassie Kennedy.

**Data curation:** Ann M. Rusk.

**Funding acquisition:** Ann M. Rusk, Victor E. Ortega, Cassie Kennedy.

**Investigation:** Ann M. Rusk, Maggie Paul, Jon Tilburt, Matthew Rank, Trudie Jackson, Alanna M. Chamberlain, Cassie Kennedy.

**Methodology:** Ann M. Rusk, Maggie Paul, Dan P. Kelleher, Jon Tilburt, Donald Northfelt, Matthew Rank, Rodrigo Cartin-Ceba, Guthrie Capossela, Trudie Jackson, Corinna Sabaque, Alanna M. Chamberlain, Victor E. Ortega, Roberto Benzo, Cassie Kennedy.

**Project administration:** Ann M. Rusk, Cassie Kennedy.

**Resources:** Ann M. Rusk, Guthrie Capossela, Trudie Jackson, Corinna Sabaque, Cassie Kennedy.

**Supervision:** Ann M. Rusk, Maggie Paul, Matthew Rank, Alanna M. Chamberlain, Cassie Kennedy.

**Validation:** Ann M. Rusk, Dan P. Kelleher.

**Visualization:** Ann M. Rusk, Alanna M. Chamberlain, Cassie Kennedy.

**Writing – original draft:** Ann M. Rusk.

**Writing – review & editing:** Ann M. Rusk, Maggie Paul, Dan P. Kelleher, Jon Tilburt, Donald Northfelt, Matthew Rank, Rodrigo Cartin-Ceba, Guthrie Capossela, Trudie Jackson, Corinna Sabaque, Alanna M. Chamberlain, Victor E. Ortega, Roberto Benzo, Cassie Kennedy.

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
