## [Decision Letter · Decision Letter 0]

28 Feb 2024

PONE-D-24-01935Identifying pragmatic solutions to reduce cigarette smoking prevalence in Indigenous North Americans: a sequential exploratory mixed-methods study protocolPLOS ONE

Dear Dr. Rusk,

Thank you for submitting your manuscript to PLOS ONE. After careful consideration, we feel that it has merit but does not fully meet PLOS ONE’s publication criteria as it currently stands. Therefore, we invite you to submit a revised version of the manuscript that addresses the points raised during the review process.

**ACADEMIC EDITOR:** ** The authors have submitted a great study which has significant positives and merits publication. However, reviewer 1 has made important comments that need to be redressed in order for us  to further consideration for submission. The authors are hence encouraged to make edits/address comments and resubmit the manuscript for further publication.**

We look forward to receiving your revised manuscript.

Kind regards,

Souparno Mitra, M.D.

Academic Editor

PLOS ONE

3. In the ethics statement in the Methods, you have specified that verbal consent was obtained. Please provide additional details regarding how this consent was documented and witnessed, and state whether this was approved by the IRB.

“Robert D. and Patricia E. Kern Center for the Science of Healthcare Delivery, Robert A. Winn Career Development Award, American Thoracic Society Fellowship in Health Equity and Diversity.”

“The authors would like to acknowledge and thank the generous funders supporting this study, including the Robert D. and Patricia E. Kern Center for the Science of Healthcare Delivery; The American Thoracic Society; The Robert A. Winn Career Development Award; the Mayo Clinic Center for Clinical and Translational Science. The authors would also like to thank the ongoing input and participation from community members who have participated in Community Engagement Studios, advice from the Healthy Nations Advisory Board, and the input from the Mayo Clinic Centers for Health Equity and Community Engagement Research in Rochester, MN, and Phoenix, AZ, for the support of this study.”

“Robert D. and Patricia E. Kern Center for the Science of Healthcare Delivery, Robert A. Winn Career Development Award, American Thoracic Society Fellowship in Health Equity and Diversity.”

7. Your ethics statement should only appear in the Methods section of your manuscript. If your ethics statement is written in any section besides the Methods, please delete it from any other section.

8. Please include a separate caption for each figure in your manuscript.

Reviewers' comments:

Reviewer's Responses to Questions

**Comments to the Author**

1. Does the manuscript provide a valid rationale for the proposed study, with clearly identified and justified research questions?

Reviewer #1: Yes

Reviewer #2: Yes

2. Is the protocol technically sound and planned in a manner that will lead to a meaningful outcome and allow testing the stated hypotheses?

Reviewer #1: Yes

Reviewer #2: Yes

3. Is the methodology feasible and described in sufficient detail to allow the work to be replicable?

Reviewer #1: Yes

Reviewer #2: Yes

4. Have the authors described where all data underlying the findings will be made available when the study is complete?

Reviewer #1: Yes

Reviewer #2: Yes

5. Is the manuscript presented in an intelligible fashion and written in standard English?

Reviewer #1: Yes

Reviewer #2: Yes

6. Review Comments to the Author

You may also provide optional suggestions and comments to authors that they might find helpful in planning their study.

Reviewer #1: The mixed-methods approach's comprehensive nature is indeed a valid point; integrating both qualitative and quantitative research provides a richer, more nuanced understanding of the targeted behaviors and experiences. The paper rightly suggests that this methodological synergy can enhance the depth and breadth of the findings, offering a more holistic perspective that pure quantitative or qualitative studies might not achieve. However, the argument's strength is directly related to the implementation quality. Simply having a mixed-methods study does not automatically translate to comprehensive insights. The concrete integration of these methods, data triangulation, and how the results will complement each other to inform the intervention are critical considerations.

The paper should clarify how the discrete choice experiments are designed, ensuring their scenarios rightly reflect realistic smoking cessation options, and that participants understand and can relate to the choices presented. The validity of the method hinges upon the extent to which these experiments mimic real-life decision-making contexts. Without this assurance, there's a valid point about the potential for compromised validity of the findings.

The localization of the study in Olmsted County, Minnesota, does raise concerns about how well the results can be extrapolated to AI/AN populations elsewhere. This limitation is important, as it affects the external validity and applicability of the results

While the paper may not address the long-term sustainability of the interventions developed, this presents a valid criticism that future research should consider. Sustainability is crucial for the long-term success of any health intervention, and the paper could improve by discussing potential strategies to maintain and scale the interventions beyond the study's scope.

Overall, the paper's mixed-methods approach, community engagement, focus on relapse prevention, and smoking during pregnancy provide strong arguments for the development of effective smoking cessation interventions in AI/AN communities. However, concerns regarding the sample size, validity of discrete choice experiments, generalizability, ethical considerations, and sustainability require further elaboration to establish the full strength and applicability of these arguments.

Reviewer #2: Overall, this is a clear, and well-written manuscript. The introduction is relevant. This is an interesting study and the data will be informative. The study design is interesting. The methodology is feasible. The manuscript is structured. It is appropriate in length.

7. PLOS authors have the option to publish the peer review history of their article (what does this mean?). If published, this will include your full peer review and any attached files.

Reviewer #1: No

Reviewer #2: No

---

## [Author Response · Author response to Decision Letter 0]

6 May 2024

Response to Reviewers:

4/1/2024

The authors would like to thank the PLOS ONE editorial team and reviewers for the thoughtful and detailed review of this protocol paper. We have conducted edits as highlighted below, and there are responses in two sections below, first, in “Journal Requirements” and second, in “Reviewers’ Comments.” Please note my comments are italicized in sections requiring a response for ease of reading. Please do not hesitate to contact me regarding additional comments, questions, or concerns regarding this manuscript.

Best, 

Annie Rusk

Journal Requirements:

The headings, level 2 and level 3 headings have been modified to match formatting requirements. Bold had been removed from figure labels. See tracked changes.

Title page and author affiliations have been changed to match journal formatting requirements. See tracked changes.

Please see completed inclusivity in global research questionnaire in submitted documents. 

3. In the ethics statement in the Methods, you have specified that verbal consent was obtained. Please provide additional details regarding how this consent was documented and witnessed, and state whether this was approved by the IRB.

The authors appreciate this comment, and have clarified the methods of consent, which can now be found under subheading “consent” following the “Indigenous Research Considerations and Community Engagement Plan”. Consent strategies are described, which include review of verbal and electronic scripts, as well as repeat of information immediately before interviews. Survey consent may occur in multiple methods, which has been highlighted in the methods section. The IRB approved materials have also been included as “Appendix 1”. 

Please see updated funding information in manuscript and disclosures. 

“Robert D. and Patricia E. Kern Center for the Science of Healthcare Delivery, Robert A. Winn Career Development Award, American Thoracic Society Fellowship in Health Equity and Diversity.”

The role of funder did not include design, data collection or analysis, publication, or manuscript preparation or production, and this statement has been added to the cover letter. 

“The authors would like to acknowledge and thank the generous funders supporting this study, including the Robert D. and Patricia E. Kern Center for the Science of Healthcare Delivery; The American Thoracic Society; The Robert A. Winn Career Development Award; the Mayo Clinic Center for Clinical and Translational Science. The authors would also like to thank the ongoing input and participation from community members who have participated in Community Engagement Studios, advice from the Healthy Nations Advisory Board, and the input from the Mayo Clinic Centers for Health Equity and Community Engagement Research in Rochester, MN, and Phoenix, AZ, for the support of this study.”

The authors thank the reviewers for clarifying where funding information should appear. The acknowledgement section has been adjusted accordingly, please see tracked changes for updates and details. 

“Robert D. and Patricia E. Kern Center for the Science of Healthcare Delivery, Robert A. Winn Career Development Award, American Thoracic Society Fellowship in Health Equity and Diversity.”

Funding information has been removed from the manuscript, the above statement is correct. 

See updated cover letter, thank you for clarification. 

7. Your ethics statement should only appear in the Methods section of your manuscript. If your ethics statement is written in any section besides the Methods, please delete it from any other section.

Sentence regarding ethics has been removed from “Conclusions” section, see tracked changes.

8. Please include a separate caption for each figure in your manuscript.

Figures have been updated to match above guidelines and descriptions/captions added to figure files.

The appendices have been renamed in the text and in file nomenclature to match the requirements of the journal. 

I have reviewed citations with tools from my citation manager software and have not identified any retractions. 

Reviewers’ Comments

1. Does the manuscript provide a valid rationale for the proposed study, with clearly identified and justified research questions?

Reviewer #1: Yes

Reviewer #2: Yes

2. Is the protocol technically sound and planned in a manner that will lead to a meaningful outcome and allow testing the stated hypotheses?

Reviewer #1: Yes

Reviewer #2: Yes

3. Is the methodology feasible and described in sufficient detail to allow the work to be replicable?

Reviewer #1: Yes

Reviewer #2: Yes

4. Have the authors described where all data underlying the findings will be made available when the study is complete?

Reviewer #1: Yes

Reviewer #2: Yes

5. Is the manuscript presented in an intelligible fashion and written in standard English?

Reviewer #1: Yes

Reviewer #2: Yes

6. Review Comments to the Author

You may also provide optional suggestions and comments to authors that they might find helpful in planning their study.

Reviewer #1: The mixed-methods approach's comprehensive nature is indeed a valid point; integrating both qualitative and quantitative research provides a richer, more nuanced understanding of the targeted behaviors and experiences. The paper rightly suggests that this methodological synergy can enhance the depth and breadth of the findings, offering a more holistic perspective that pure quantitative or qualitative studies might not achieve. However, the argument's strength is directly related to the implementation quality. Simply having a mixed-methods study does not automatically translate to comprehensive insights. The concrete integration of these methods, data triangulation, and how the results will complement each other to inform the intervention are critical considerations.

The paper should clarify how the discrete choice experiments are designed, ensuring their scenarios rightly reflect realistic smoking cessation options, and that participants understand and can relate to the choices presented. The validity of the method hinges upon the extent to which these experiments mimic real-life decision-making contexts. Without this assurance, there's a valid point about the potential for compromised validity of the findings.

The localization of the study in Olmsted County, Minnesota, does raise concerns about how well the results can be extrapolated to AI/AN populations elsewhere. This limitation is important, as it affects the external validity and applicability of the results

While the paper may not address the long-term sustainability of the interventions developed, this presents a valid criticism that future research should consider. Sustainability is crucial for the long-term success of any health intervention, and the paper could improve by discussing potential strategies to maintain and scale the interventions beyond the study's scope.

Overall, the paper's mixed-methods approach, community engagement, focus on relapse prevention, and smoking during pregnancy provide strong arguments for the development of effective smoking cessation interventions in AI/AN communities. However, concerns regarding the sample size, validity of discrete choice experiments, generalizability, ethical considerations, and sustainability require further elaboration to establish the full strength and applicability of these arguments.

The authors acknowledge the challenges of implementing interventions in diverse systems with variable resources. These limitations may be profound, and edits have been added in the tracked changes of the document in the discussion. Further description of validation of the DCE survey has also been added to the methods. The authors appreciate this thoughtful comment highlighting the importance of acknowledging the diversity that exists among AI/AN people in the US.

Reviewer #2: Overall, this is a clear, and well-written manuscript. The introduction is relevant. This is an interesting study and the data will be informative. The study design is interesting. The methodology is feasible. The manuscript is structured. It is appropriate in length.

7. PLOS authors have the option to publish the peer review history of their article (what does this mean?). If published, this will include your full peer review and any attached files.

Do you want your identity to be public for this peer review? For information about this choice, including consent withdrawal, please see our Privacy Policy.

Reviewer #1: No

Reviewer #2: No

---

## [Decision Letter · Decision Letter 1]

20 Jun 2024

Identifying pragmatic solutions to reduce cigarette smoking prevalence in Indigenous North Americans: a sequential exploratory mixed-methods study protocol

PONE-D-24-01935R1

Dear Dr. Rusk,

We’re pleased to inform you that your manuscript has been judged scientifically suitable for publication and will be formally accepted for publication once it meets all outstanding technical requirements.

Kind regards,

Souparno Mitra, M.D.

Academic Editor

PLOS ONE

Additional Editor Comments (optional):

Reviewers' comments:

Reviewer's Responses to Questions

**Comments to the Author**

1. Does the manuscript provide a valid rationale for the proposed study, with clearly identified and justified research questions?

Reviewer #2: Yes

Reviewer #3: Yes

2. Is the protocol technically sound and planned in a manner that will lead to a meaningful outcome and allow testing the stated hypotheses?

Reviewer #2: Yes

Reviewer #3: Yes

3. Is the methodology feasible and described in sufficient detail to allow the work to be replicable?

Reviewer #2: Yes

Reviewer #3: Yes

4. Have the authors described where all data underlying the findings will be made available when the study is complete?

Reviewer #2: Yes

Reviewer #3: Yes

5. Is the manuscript presented in an intelligible fashion and written in standard English?

Reviewer #2: Yes

Reviewer #3: Yes

6. Review Comments to the Author

You may also provide optional suggestions and comments to authors that they might find helpful in planning their study.

Reviewer #2: Overall, this is a clear, and well-written manuscript. The introduction is relevant. This is an interesting study and the data will be informative. The study design is interesting. The methodology is feasible. The manuscript is structured. It is appropriate in length.

Reviewer #3: The proposed study design in the article seems strong and well-suited to tackle the complex issue of smoking cessation in AI/AN populations. It shows a dedication to involving the community, using rigorous methodology, and ensuring practical relevance, all of which are crucial for producing impactful research results. The focus on not only identifying the reasons for smoking prevalence but also defining effective solutions is commendable, especially the practical approach that increases the likelihood of interventions being culturally appropriate, practical, and implementable. Some suggestions for improvement, such as clarifying the experimental design and addressing concerns about generalizability and sustainability, are well-founded and would enhance the paper's quality.

7. PLOS authors have the option to publish the peer review history of their article (what does this mean?). If published, this will include your full peer review and any attached files.

Reviewer #2: No

Reviewer #3: No

---

## [Editor Report · Acceptance letter]

10 Sep 2024

PONE-D-24-01935R1 

PLOS ONE

Dear Dr. Rusk, 

I'm pleased to inform you that your manuscript has been deemed suitable for publication in PLOS ONE. Congratulations! Your manuscript is now being handed over to our production team.

Kind regards, 

on behalf of

Dr. Souparno Mitra 

Academic Editor

PLOS ONE